# Effects of Melatonin and 3,5,3′-Triiodothyronine on the Development of Rat Granulosa Cells

**DOI:** 10.3390/nu16183085

**Published:** 2024-09-13

**Authors:** Mingqi Wu, Yilin Yao, Rui Chen, Baoqiang Fu, Ying Sun, Yakun Yu, Yan Liu, Haoyuan Feng, Shuaitian Guo, Yanzhou Yang, Cheng Zhang

**Affiliations:** 1College of Life Science, Capital Normal University, Beijing 100048, China; 2220802057@cnu.edu.cn (M.W.); 2230801011@cnu.edu.cn (Y.Y.); 2230802062@cnu.edu.cn (R.C.); bejayfu@163.com (B.F.); 2220802055@cnu.edu.cn (Y.S.); 2230802063@cnu.edu.cn (H.F.); 2230802064@cnu.edu.cn (S.G.); 2Key Laboratory of Fertility Preservation and Maintenance, Ministry of Education, Key Laboratory of Reproduction and Genetics in Ningxia, Department of Histology and Embryology, Ningxia Medical University, Yinchuan 750004, China

**Keywords:** melatonin, T_3_, H_2_O_2_, GRP78, AMPK, SIRT1, granulosa cells

## Abstract

Melatonin, as an endocrine neurotransmitter, can promote the development of the ovary. Meanwhile, it also has protective effect on the ovary as an antioxidant. Thyroid hormone (TH) is essential for normal human reproductive function. Many studies have shown that 3,5,3′-triiodothyronine (T_3_) regulates the development of ovarian granulosa cells. However, little is known about the specific mechanisms by which melatonin combines with T_3_ to regulate granulosa cell development. The aim of present study was to investigate the effects and the possible mechanisms of melatonin and T_3_ on ovarian granulosa cell development. In the present study, cell development and apoptosis were detected by CCK8, EdU and TUNEL, respectively. The levels of related proteins were analyzed by Western blotting. The results showed that oxidative stress (OS) and reactive oxygen species (ROS) were induced by H_2_O_2_ in granulosa cells, and cell apoptosis was also increased accompanied with the decreased cellular proliferation and viability. Melatonin protects granulosa cells from H_2_O_2_-induced apoptosis and OS by downregulating ROS levels, especially in the presence of T_3_. Co-treatment of cell with melatonin and T_3_ also promotes the expression of GRP78 and AMH, while inhibiting CHOP, Caspase-3, and P16. It was demonstrated that melatonin alone or in combination with T_3_ had positive effect on the development of granulosa cells. In addition, the AMPK/SIRT1 signaling pathway is involved in the process of melatonin/T_3_ promoting granulosa cell development.

## 1. Introduction

Mammalian follicular development from the primordial follicle to ovulation is a highly complex and tightly regulated process [1,2], which is also accompanied by the follicle atresia [3]. Granulosa cells are crucial for the follicular development, and their apoptosis is closely related to follicle atresia [4,5,6,7].

It is well known that incomplete reduction of oxygen leads to the generation of reactive oxygen species (ROS), causing oxidative stress (OS) and affecting normal cell function [8]. Studies have shown that H_2_O_2_ can induce granulosa cell apoptosis through the ROS-JNK-p53 pathway [9]. OS has significant effects on mitochondrial activity, proliferation, differentiation and cell cycle in granulosa cells [10]. Endoplasmic reticulum stress (ERS) also affects ovarian granulosa cell development. The excessive ERS may trigger cell apoptosis by regulating the expression of CHOP and caspase-3 [11]. In goat follicles, ERS-related apoptotic proteins are increased with the upregulation of GRP78 (Glucose-regulated protein 78), which plays many important roles in cell survival and apoptosis [12]. Increased evidence has shown that OS and ERS interact with each other to regulate cellular development. Ovarian dysfunction in patients with endometriosis is associated with ERS combined with OS by regulating granulosa cell apoptosis [13]. In neuronal cells, excessive OS induces ERS and then increases cell apoptosis [14]. However, the regulation of ovarian granulosa cells by OS and ERS still remains to be explored.

It has been reported that melatonin can act against OS and regulate ovarian development [15]. A previous study has shown that melatonin has a positive effect on granulosa cells and stimulates the synthesis of estradiol [16]. Moreover, melatonin can interact with thyroid hormone (TH) to regulate ovarian functions. It has been reported that dysregulation of TH impairs ovarian function, and TH indirectly or directly regulates follicular development [17,18]. Our previous studies have shown that 3,5,3′-triiodothyronine (T_3_) combined with follicle-stimulating hormone (FSH) inhibits cell apoptosis and promotes cell proliferation by increasing steroid hormone synthesis and glucose uptake in granulosa cells [11,19,20,21]. Although many studies have reported that melatonin and TH are important in granulosa cell development, the exact mechanisms by which melatonin and T_3_ regulate ovarian granulosa cell development are still unclear.

Anti-Mullerian hormone (AMH) is a reproductive hormone representing ovarian reserve capacity, which inhibits follicle recruitment and affects cells normal growth and differentiation. Lower concentration of AMH and impaired oocyte quality during female aging are due to the increased cellular OS and DNA damage [22]. Multiple tumor suppressor 1 (P16/MTS) inhibits cells division and is considered a marker of tissue and cellular senescence. P16 expression is related to OS and is increased by the high OS levels in the ovary, thereby inducing ovarian aging [23,24,25]. Accumulating studies suggest that OS contributes to cellular senescence; however, the mechanisms underlying the potential alleviating effects of melatonin and T_3_ on this phenomenon remain unknown.

The Adenosine 5′-monophosphate (AMP)-activated protein kinase (AMPK)/Sirtuin1 (SIRT1) pathway is a well-established important biological metabolic pathway that regulates many processes in numerous cells. Moreover, AMPK and SIRT1 also interact with each other and participate in cell aging, life span regulation, and other functions. The SIRT1/AMPK axis is involved in autophagy activation and promotes ovarian development [26]. The AMPK/SIRT1 signaling pathway also plays a key role in delaying premature ovarian failure (POF) [27]. The regulation of AMPK/SIRT1 is diverse and complex. However, whether and how AMPK and SIRT1 modulate follicular development is not known.

In this study, we found that melatonin and T_3_ play positive roles in the development of granulosa cells by reducing OS and ERS. Moreover, the AMPK/SIRT1 signaling pathway is involved in these regulations.

## 2. Materials and Methods

### 2.1. Reagents and Antibodies

Unless otherwise specified, most of the reagents and chemicals used in the present study were purchased from Sigma-Aldrich (St. Louis, MO, USA). Rabbit polyclonal anti-AMH (ab229212), rabbit monoclonal anti-SIRT1 (ab189494), and rabbit monoclonal anti-AMPK (ab32047) were purchased from Abcam (Cambridge, MA, USA). Rabbit polyclonal anti-GRP78 (sc-13968), mouse monoclonal anti-CHOP (sc-7351), rabbit polyclonal anti-Caspase-3 (sc-7148), and mouse monoclonal anti-P16 (sc-1661) were purchased from Santa Cruz Biotechnology, Inc. (Dallas, TX, USA). Rabbit monoclonal anti-p-AMPK (#2537) was obtained from Cell Signaling Technology, Inc. (Danvers, MA, USA). Rabbit polyclonal anti-β-actin (BE39995), horse radish peroxidase (HRP)-conjugated anti-rabbit and anti-mouse IgG were from Bio-easy (Beijing, China). The inhibitors of AMPK (Compound C) and SIRT1 (EX527) were purchased from Selleck (Houston, TX, USA). Medium 199 (M199) was purchased from Biological Industries (Beit Haemek, Israel). Fetal bovine serum was from PAN-Biotech (Bavaria, Aidenbach, Germany). The penicillin and streptomycin were purchased from Invitrogen (Carlsbad, CA, USA). The enhanced chemiluminescence (ECL) detection kit was obtained from LABLEAD (Beijing, China).

### 2.2. Animal Treatments

The 21-day-old Sprague Dawley (SD) female rats were purchased from the Beijing Vital Laboratory Animal Technology (Beijing, China) and kept at constant temperature (24–26 °C) and humidity (60% ± 2%) with a light/dark cycle of 12/12 h. Ovaries were collected after administration of diethylstilbestrol (DES) (10 mg/mL) to immature SD rats by intraperitoneal injection for three consecutive days [28]. All animals were euthanized using carbon dioxide inhalation. All procedures were approved by the Institutional Animal Care and Use Committee of Capital Normal University and conducted in accordance with the Principles of the Care and Use of Laboratory Animals and China Council on Animal Care.

### 2.3. Rat Granulosa Cell Isolation and Culture

The processes of granulosa cell isolation and culture were performed as described before [29]. The culture medium was prepared using M199 culture medium and supplemented with 10% fetal bovine serum and 1% antibiotics. The cells were maintained under a humidified atmosphere of 95% air and 5% CO_2_ at 37 °C. Briefly, the ovaries were punctured with a 1 mL syringe needle under a stereoscope, and then the cell suspensions were filtered through a nylon cell strainer. Approximately 9 × 10^5^ viable granulosa cells were isolated from 2–3 ovaries in each group (per well in a 6-well plate). The cells were centrifuged and subsequently plated in pre-equilibrated culture medium, followed by 24 h of adhesion culture. Then, cells were pretreated with H_2_O_2_ for 1 h and subsequently cultured with melatonin (0.1 μM) (Lot # WXBC9956V, purity 99%) and/or T_3_ (1.0 nM) (Lot # BCCC9589, purity 99%) for 48 h. Some groups were treated with melatonin (0.1 μM) and T_3_ (1.0 nM) for 0 min, 30 min, 1 h, 6 h, 12 h or 24 h. For some experiments, cells were pretreated with AMPK or SIRT1 inhibitors (10 μM) for 24 h prior to being cultured with melatonin (0.1 μM) and T_3_ (1.0 nM) for 48 h.

### 2.4. Protein Extraction and Western Blotting

Western blotting analysis was performed as described previously [30]. The cell pellets were lysed by incubation with lysis buffer at 0 °C for 30 min, and the supernatant was collected for experiments after centrifugation (15,000× *g*, 4 °C, 30 min). Protein concentration was determined by the BCA Protein Assay kit (Beyotime Biotechnology, Shanghai, China). Total proteins (20 µg) were separated by SDS-PAGE and transferred to polyvinylidene difluoride (PVDF) membranes (Roche, Basel, Switzerland). Subsequently, the membranes were incubated with 5% bovine serum protein for 1 h at room temperature, and then incubated with diluted primary antibody [polyclonal anti-AMH (1:500), monoclonal anti-AMPK (1:500), monoclonal anti-p-AMPK (1:500), monoclonal anti-CHOP (1:500), polyclonal anti-Caspase-3 (1:500), polyclonal anti-GRP78 (1:1000), monoclonal anti-P16 (1:200), monoclonal anti-SIRT1 (1:500), or polyclonal anti-β-actin (1:500)] overnight at 4 °C, followed by incubation with HRP-conjugated secondary antibody (1:2000–1:5000) for 2 h at room temperature. Peroxidase activity was observed and visualized with the ECL kit and ImageQuant LAS 4000 Mini imaging system (Cytiva, Marlborough, MA, USA) according to the manufacturer’s instructions. The immune response signals were analyzed using AlphaEaseFC 4.0 (Alpha Innotech, San Leandro, CA, USA).

### 2.5. Detection of ROS

In accordance with instructions, DCFH-DA (1:1000) was diluted to a final concentration of 10 μM. After the treatment drug was removed, the cells were washed 3 times with 1× phosphate-buffered saline (1×PBS), and then diluted DCFH-DA solution was added to fully cover all the cells. Subsequently, the cells were incubated at 37 °C for 30 min. The adherent growth cells were examined under a fluorescence microscope and photographed. The experimental results were analyzed using ImageJ 1.53e.

### 2.6. Analysis of Cell Viability

CCK-8 (Dojindo, Kumamoto, Japan) was used to detect the cell viability, as described previously [20]. The cells were incubated with 10 µL CCK-8 solution for 2–3 h at 37 °C, and optical density values were measured by a microplate reader at 450 nm. The mean optical density value of each treatment was used as an index of cell viability.

### 2.7. EdU Incorporation Assay

EdU cell proliferation assays were performed according to the manufacturer’s instructions (Beyotime Biotechnology, Shanghai, China). Briefly, cells were cultured and incubated with 50 µM EdU for 2 h, followed by three washes in 1×PBS. After fixation with 4% paraformaldehyde for 30 min at room temperature and permeabilization with 0.5% Triton X-100 for 10 min, cells were stained for EdU. Cell nuclei were counterstained with Hoechst 33342 for 30 min. EdU-positive nuclei were detected using a laser scanning microscope, LSM 780 (Zeiss, Jena, Germany). The proportion of nucleated cells containing EdU in five high-power fields/well was used to calculate the rate of cell proliferation.

### 2.8. TUNEL Analysis

Apoptotic cells were identified using a TUNEL Apoptosis Assay Kit (KeyGEN, Beijing, China). In brief, cells were fixed with 4% paraformaldehyde for 30 min at room temperature and then permeated with 1% Triton X-100 for 5 min. After washing with 1×PBS, cells were incubated in TdT buffer for 1 h at 37 °C in a humidified chamber. Then, 50 µL of the Streptavidin-TRITC labeling solution was added to each well and incubated in the dark at 37 °C for 30 min. To stain the nuclei, cells were incubated with diluted DAPI solution (1:200) for 10 min at room temperature in dark. Subsequently, the cells were photographed with a laser scanning microscope, LSM 780 (Zeiss, Jena, Germany), and the images were recorded using Zeiss ZEN lite software (ZEN 2, blue edition).

### 2.9. Statistical Analysis

The experiments were repeated at least three times. All experimental data are presented as mean ± SEM. The data were analyzed by GraphPad Prism 8.3.0 software (GraphPad Software, La Jolla, CA, USA) and the statistical differences between groups were calculated with a *t*-test and one-way (repeated-measures) ANOVA. Means were compared by the Bonferroni post-test when significant differences were found. Additionally, statistically significant differences were considered at *p* < 0.05.

## 3. Results

### 3.1. Effect of H_2_O_2_ on Granulosa Cells

To explore the effect of OS on the development of granulosa cells, cells were treated by H_2_O_2_ with different concentrations (0, 100, 200, 400 and 800 μM) for 1 h, and cell viability was determined by CCK8 assay. As shown in Figure 1, cell viability was significantly decreased by H_2_O_2_ in a dose–response manner (*p* < 0.0001), which suggests that cell growth was affected by higher OS.

### 3.2. Effect of Melatonin/T_3_ on H_2_O_2_-Induced ROS in Granulosa Cells

To explore the effect of melatonin/T_3_ on OS in granulosa cells, cells were pre-treated with H_2_O_2_ for 1 h, and then treated with melatonin (0.1 μM) and/or T_3_ (1.0 nM) for 48 h. The concentrations of melatonin and T_3_ were selected based on the preliminary experiments (Appendix A). The fluorescence intensity of DCF was detected, and the content of ROS was measured. As shown in Figure 2, ROS levels were significantly increased by H_2_O_2_ (*p* < 0.0001). However, this induction was dramatically reduced by melatonin and T_3_ alone (*p* < 0.001). Moreover, the combination of melatonin and T_3_ was more effective in reducing ROS than hormone alone (*p* < 0.01).

### 3.3. Effects of Melatonin/T_3_ on Cell Growth of H_2_O_2_-Treated Granulosa Cells

To demonstrate the effect of melatonin/T_3_ on cellular development in rat granulosa cells under OS, cell viability and proliferation were investigated after hormone treatment. As shown in Figure 3, granulosa cell viability was significantly reduced by H_2_O_2_ (*p* < 0.01, Figure 3A), and the decreasing effect was reversed by melatonin, T_3_ (*p* < 0.01, *p* < 0.001, Figure 3A), and especially by the combination of these hormones (*p* < 0.05, Figure 3A).

Additionally, H_2_O_2_’s inhibitory effect on cell proliferation was significantly alleviated by melatonin, T_3_ treatment (*p* < 0.001, Figure 3B,C), and especially by the combination of melatonin and T_3_ (*p* < 0.05, Figure 3B,C).

### 3.4. Effect of Hormones on Granulosa Cells’ Apoptosis

To explore the effect of melatonin/T_3_ on cellular apoptosis, granulosa cells were induced by H_2_O_2_ before hormone treatment. As shown in Figure 4, H_2_O_2_ significantly increased granulosa cell apoptosis (*p* < 0.001). However, H_2_O_2_-induced apoptosis was significantly inhibited by melatonin (*p* < 0.01) and T_3_ (*p* < 0.01) alone. The combined hormone treatment showed the most pronounced effect.

### 3.5. Effects of Melatonin/T_3_ on the Expression of AMH and P16 in Granulosa Cells

It is well known that AMH is produced by the granulosa cells of small, growing follicles, which are regarded as the marker of ovarian reserve. In order to investigate the effects of hormones on AMH expression, cells were pretreated with H_2_O_2_ before hormone treatment. The results showed that H_2_O_2_ treatment significantly suppressed AMH expression (*p* < 0.05, Figure 5A). This suppression was effectively countered by subsequent treatment with either melatonin or T_3_ (*p* < 0.05, *p* < 0.01, Figure 5A), and especially by combined treatment with both hormones (*p* < 0.05, Figure 5A).

In addition, H_2_O_2_ induced a significant upregulation of P16 (*p* < 0.01, Figure 5B), which was abrogated by treatment with melatonin and/or T_3_ (*p* < 0.01, *p* < 0.001, Figure 5B), with the most significant reduction observed in the combined treatment (*p* < 0.01, Figure 5B).

### 3.6. Effects of Melatonin/T_3_ on ERS in Granulosa Cells

To further detect the effect of H_2_O_2_ on ERS in granulosa cells, the expression of ERS-related proteins, GRP78, CHOP, and Caspase-3, were investigated. The results showed that H_2_O_2_ significantly downregulated the expression of the ERS-associated protein GRP78 (*p* < 0.01, Figure 6A), and the inhibitory effect was mitigated by individual treatments with melatonin or T_3_ alone (*p* < 0.05, *p* < 0.01, Figure 6A). Notably, co-treatment with both hormones further increased GRP78 expression (*p* < 0.05, Figure 6A).

Meanwhile, H_2_O_2_-induced upregulations of CHOP and Caspase-3 were significantly reversed by melatonin or T_3_ treatments (*p* < 0.01, Figure 6B,C), with the combination of both hormones showing a more pronounced inhibitory effect (*p* < 0.05, Figure 6B).

### 3.7. Effect of Melatonin/T_3_ on the Expression of AMPK and SIRT1 in Granulosa Cells

To investigate whether H_2_O_2_ affects the expression of AMPK and SIRT1, the protein contents were detected. H_2_O_2_ treatment significantly decreased the p-AMPK protein level (*p* < 0.05, Figure 7A), and the effect was substantially reversed by hormone treatment (*p* < 0.01, Figure 7A). Similarly, H_2_O_2_ suppressed SIRT1 expression (*p* < 0.05, Figure 7B), and this suppression was also counteracted by T_3_ (*p* < 0.05, Figure 7B), especially in the presence of melatonin (*p* < 0.05, Figure 7B).

### 3.8. The AMPK/SIRT1 Signaling Pathway Is Involved in the Regulation of Cellular Development by Melatonin/T_3_

To examine whether melatonin/T_3_ regulate AMPK and SIRT1, granulosa cells were treated with melatonin and T_3_, and then the expression levels of p-AMPK and SIRT1 were detected. The results showed that the expression level of p-AMPK peaked at 1 h after melatonin and T_3_ co-treatment (*p* < 0.05, Figure 8A), and the expression level of SIRT1 peaked at 6 h after hormone treatment (*p* < 0.01, Figure 8B).

In order to determine whether the AMPK/SIRT1 signaling pathway is involved in the regulation of cellular development, granulosa cells were pretreated with Compound C (AMPK inhibitor, 10 μM) or EX527 (SIRT1 inhibitor, 10 μM) for 24 h and then treated with melatonin and T_3_ for 48 h. As shown in Figure 8C,D, the expression of AMH and GRP78 in granulosa cells was significantly reduced by the inhibitors after melatonin and T_3_ treatment (*p* < 0.05, *p* < 0.01, Figure 8C,D).

## 4. Discussion

In the present study, we investigated the effects of melatonin and/or T_3_ on ovarian granulosa cells. Our results demonstrated that melatonin and T_3_ have protective effects against hydrogen peroxide (H_2_O_2_)-induced oxidative damage to granulosa cells, especially in the presence of both hormones. Additionally, the AMPK/SIRT1 signaling pathway is involved in this regulation.

Studies have found that OS is an imbalance between oxidative and anti-oxidative processes in vivo, which is involved in the pathogenesis of different diseases. OS also regulates the survival and development of granulosa cells [12]. In the present experiment, H_2_O_2_ induced higher levels of ROS. Additionally, cell proliferation and viability were significantly reduced with increased apoptosis, which caused negative effects on cells.

It has been reported that the occurrence of POF may be closely related to OS in the ovary [31,32]. Our results showed that H_2_O_2_ decreased the expression of AMH as a marker of ovarian aging [25]. Moreover, the expression of P16 was significantly increased in granulosa cells after H_2_O_2_ treatment. As a cell cycle suppressor gene, P16 is highly expressed in the process of inhibiting cell differentiation [28], and P16 is currently considered a tissue and cellular marker of senescence [33,34,35]. Some studies have found that the P16 protein is expressed at low levels in the tissues of young animals, and its expression increases with age [36]. These results indicate that H_2_O_2_ could induce POF by increasing cellular senescence and downregulating AMH content.

OS is also implicated in the cellular apoptosis. Autophagy is promoted by activating ROS and ERS in mouse granulosa cells [37]. Some forms of ROS can interfere with protein folding in endoplasmic reticulum and induce ERS [38]. GRP78 plays an important role in protein synthesis and is the central regulator of ERS [11]. It participates in protein assembly and folding and binds to misfolded proteins to prevent protein agglutination. GRP78 can regulate the balance between cell survival and apoptosis in ERS cells [39]. The GRP78 receptor can be detected on the surface of rat ovarian granulosa cells and regulates cell proliferation and survival [40,41]. It was found that ERS enhanced the expression of GRP78 in mouse granulosa cells and triggered cell apoptosis [42]. Some studies have also shown that ERS-related apoptotic proteins are increased with the expression of GRP78, which regulates granulosa cell apoptosis and participates in the process of follicular atresia in goats [12]. In the present study, the expression of GRP78 in granulosa cells after H_2_O_2_ treatment was significantly decreased, and the expression of CHOP and Caspase-3 were dramatically increased. However, these effects induced by H_2_O_2_ were significantly reversed by melatonin alone or in combination with T_3_. These results indicate that GRP78 may play protective roles in cellular development after H_2_O_2_ treatment.

Melatonin has extensive physiological functions in different tissues and cells. Melatonin also has protective effects on OS-induced apoptosis in different cells. The antioxidant effect of melatonin is closely related to the removal of OS [35,43]. It has been reported that melatonin is essential in reducing the production of cellular OS to promote oocyte maturation and granulosa cell development [11]. Meanwhile, melatonin is considered -an anti-aging agent based on its cytoprotective effect. In the current study, melatonin improved the development of granulosa cells. The attenuation of apoptosis by melatonin in granulosa cells might be due to decreased ROS production. Moreover, the same positive effects were also shown after T_3_ treatment in H_2_O_2_-induced granulosa cells. Interestingly, the protective effects of the combination of melatonin and T_3_ on granulosa cells were significantly higher than that of melatonin or T_3_ alone. The effects of these hormones may be due to the antioxidant and anti-aging properties [22]. In addition, melatonin is a seasonal hormone, which is regulated by photoperiod [44]. The effects of melatonin regulated by photoperiod on reproduction still need to be further investigated in the future. Whether hormones alleviate ovarian dysfunction by reducing OS in vivo remains to be explored.

It has been reported that OS is closely related to energy transfer, and AMPK is the main energy sensor for energy homeostasis in eukaryotes [45,46]. AMPK is significantly reduced when cells undergo OS [34,47], which regulates downstream pathways. SIRT1, as a downstream regulator of AMPK, plays roles in regulating gene transcription and DNA repair [19,48]. Many studies have shown that AMPK/SIRT1 plays an important role in promoting cell signal transduction, and this pathway is involved in the regulation of follicular maturation and development [49,50]. In the present study, H_2_O_2_ decreased AMPK and SIRT1 expression in granulosa cells. Moreover, the expressions of AMH and GRP78 were significantly decreased by AMPK or SIRT1 inhibitors, respectively. These results indicate that the AMPK/SIRT1 pathway is involved in the upregulation of AMH and GRP78 expression by melatonin/T_3_.

## 5. Conclusions

Our findings demonstrate that OS and ERS significantly impair the viability and developmental potential of rat granulosa cells, as evidenced by H_2_O_2_-induced apoptosis and reduced cell proliferation. Melatonin and T_3_, and particularly the combination of both hormones, have protective effects on granulosa cell development. In addition, the AMPK/SIRT1 signaling pathway is involved in the regulation of AMH and GRP78 expression in granulosa cells by melatonin and T_3_. Our results enhance our understanding of the roles of melatonin and thyroid hormone in suppressing ERS and OS to promote the development of granulosa cells. Our findings provide valuable insights into the protective mechanisms of granulosa cells and lay the groundwork for improving ovarian health and fertility.

## Figures and Tables

**Figure 1 nutrients-16-03085-f001:**
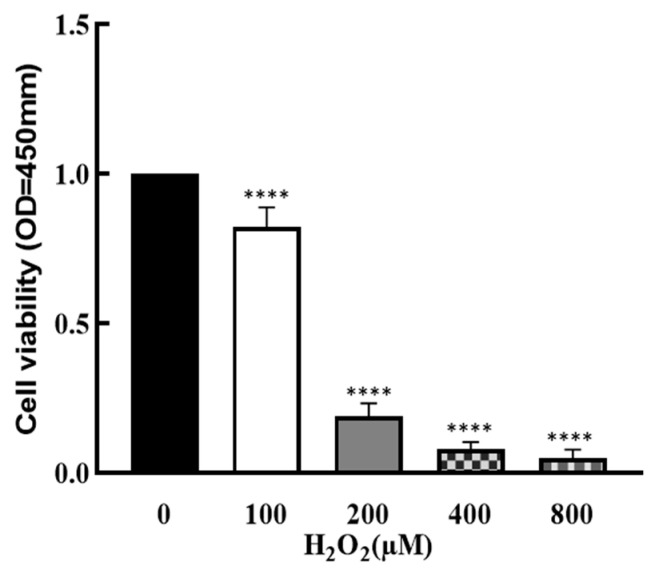
Effect of H_2_O_2_ on granulosa cell viability. The ovarian granulosa cells were pretreated with different concentrations of H_2_O_2_ (0, 100, 200, 400 and 800 μM) for 1 h, and a CCK-8 assay was used to analyze the cell viability. **** *p* < 0.0001, compared with 0 μM group.

**Figure 2 nutrients-16-03085-f002:**
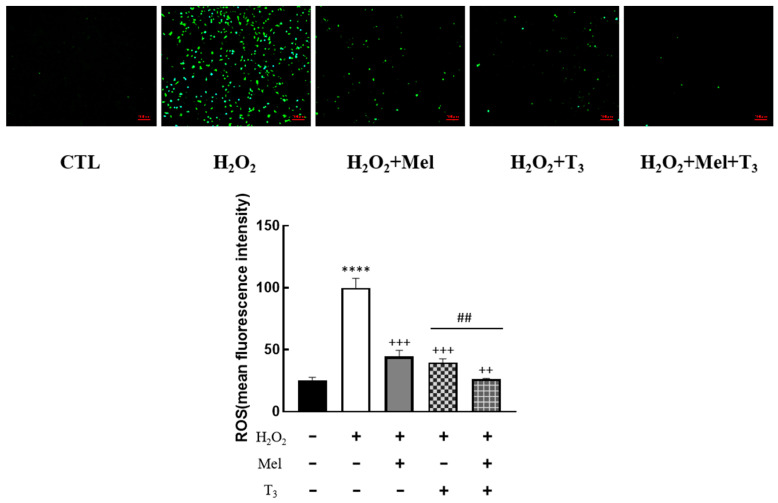
Effects of melatonin/T_3_ on ROS in rat granulosa cells. After granulosa cells were induced by H_2_O_2_ (100 μM) for 1 h, granulosa cells were cultured with melatonin (0.1 μM) and/or T_3_ (1.0 nM) for 48 h, and the ROS levels were measured. **** *p* < 0.0001, compared with CTL; ^++^ *p* < 0.01, ^+++^ *p* < 0.001, compared with H_2_O_2_; ^##^ *p* < 0.01, compared with H_2_O_2_ + T_3_. Scale bars = 100 μm.

**Figure 3 nutrients-16-03085-f003:**
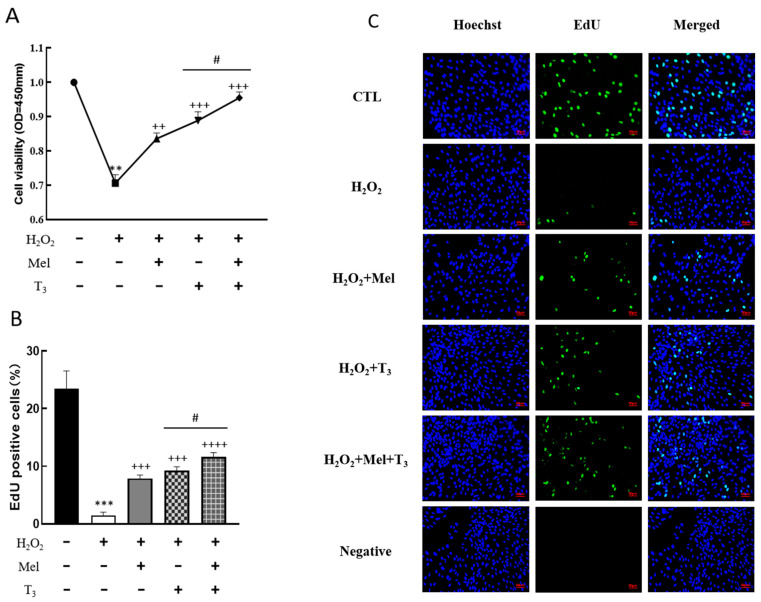
Effects of melatonin/T_3_ on the viability and proliferation of rat granulosa cells. Granulosa cells were induced by H_2_O_2_ (100 μM for 1 h) and cultured with melatonin (0.1 μM) and/or T_3_ (1.0 nM) for 48 h. Cell viability (**A**) and proliferation (**B**,**C**) were detected by CCK-8 assay and EdU, respectively. The nucleus was stained with Hoechst (blue fluorescence). Data are presented as mean ± standard error of the mean of three independent experiments. ** *p* < 0.01, *** *p* < 0.001, compared with CTL; ^++^ *p* < 0.05, ^+++^ *p* < 0.001, ^++++^ *p* < 0.0001, compared with H_2_O_2_; ^#^ *p* < 0.05, compared with H_2_O_2_ + T_3_. Scale bars = 50 μm.

**Figure 4 nutrients-16-03085-f004:**
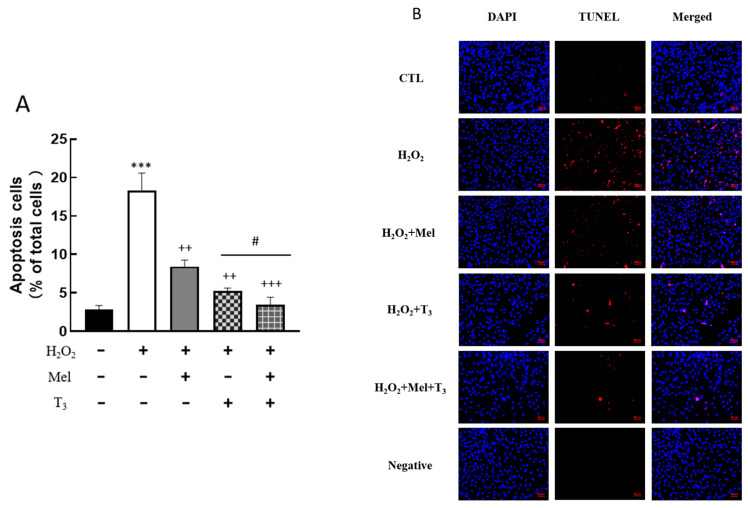
Effects of melatonin/T_3_ on apoptosis of rat granulosa cells. Granulosa cells were induced by H_2_O_2_ (100 μM for 1 h) and then cultured with melatonin (0.1 μM) and/or T_3_ (1.0 nM) for 48 h. A TUNEL assay was used to measure the apoptosis (**A**,**B**) of granulosa cells in different groups. The nucleus was stained with DAPI (blue fluorescence). *** *p* < 0.001, compared with CTL; ^++^ *p* < 0.01, ^+++^ *p* < 0.001, compared with H_2_O_2_; ^#^ *p* < 0.05, compared with H_2_O_2_ + T_3_. Scale bars = 50 μm.

**Figure 5 nutrients-16-03085-f005:**
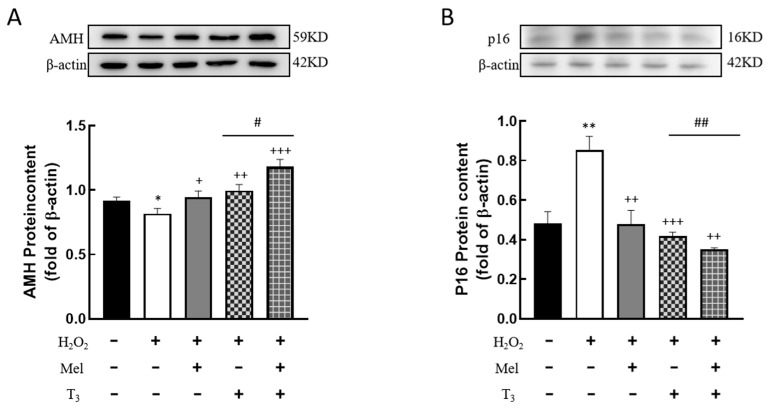
Effects of melatonin/T_3_ on AMH and P16 content in granulosa cells. The proteins expression of AMH (**A**) and P16 (**B**) were determined by Western blotting. * *p* < 0.05, ** *p* < 0.01, compared with CTL; ^+^ *p* < 0.05, ^++^ *p* < 0.01, ^+++^ *p* < 0.001, compared with H_2_O_2_; ^#^ *p* < 0.05, ^##^ *p* < 0.01, compared with H_2_O_2_ + T_3_.

**Figure 6 nutrients-16-03085-f006:**
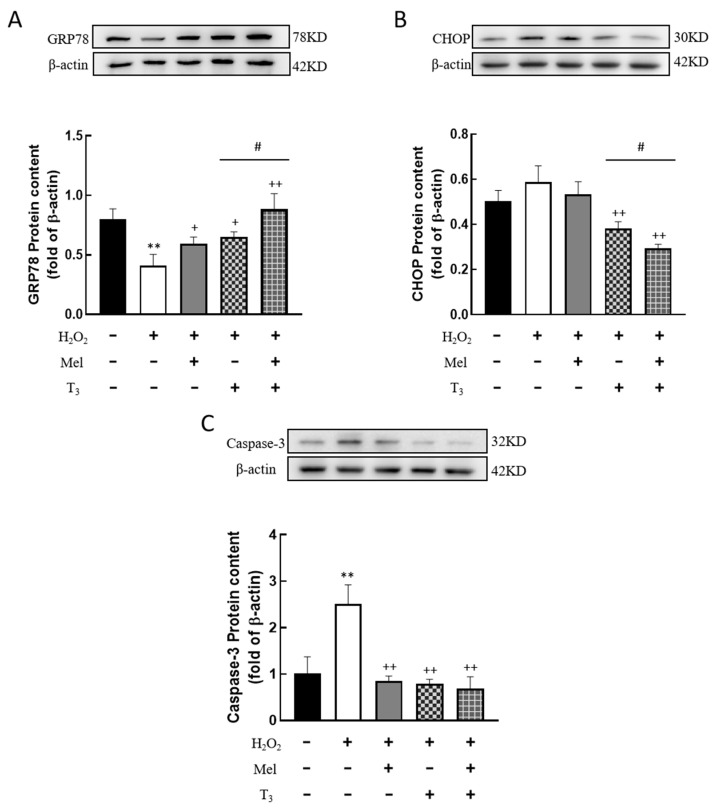
Effects of melatonin/T_3_ on ERS related proteins in granulosa cells. The protein expression of GRP78 (**A**), CHOP (**B**) and Caspase-3 (**C**) was determined by Western blotting. ** *p* < 0.01, compared with CTL; ^+^ *p* < 0.05, ^++^ *p* < 0.01, compared with H_2_O_2_; ^#^ *p* < 0.05, compared with H_2_O_2_ + T_3_.

**Figure 7 nutrients-16-03085-f007:**
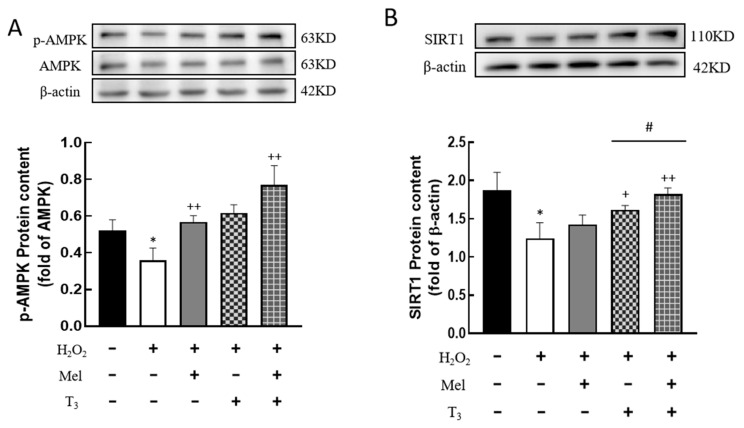
Effects of melatonin/T_3_ on AMPK and SIRT1 in granulosa cells. The protein expression of AMPK (**A**) and SIRT1 (**B**) was determined by Western blotting. * *p* < 0.05, compared with CTL; ^+^ *p* < 0.05, ^++^ *p* < 0.01, compared with H_2_O_2_; ^#^ *p* < 0.05, compared with H_2_O_2_ + T_3_.

**Figure 8 nutrients-16-03085-f008:**
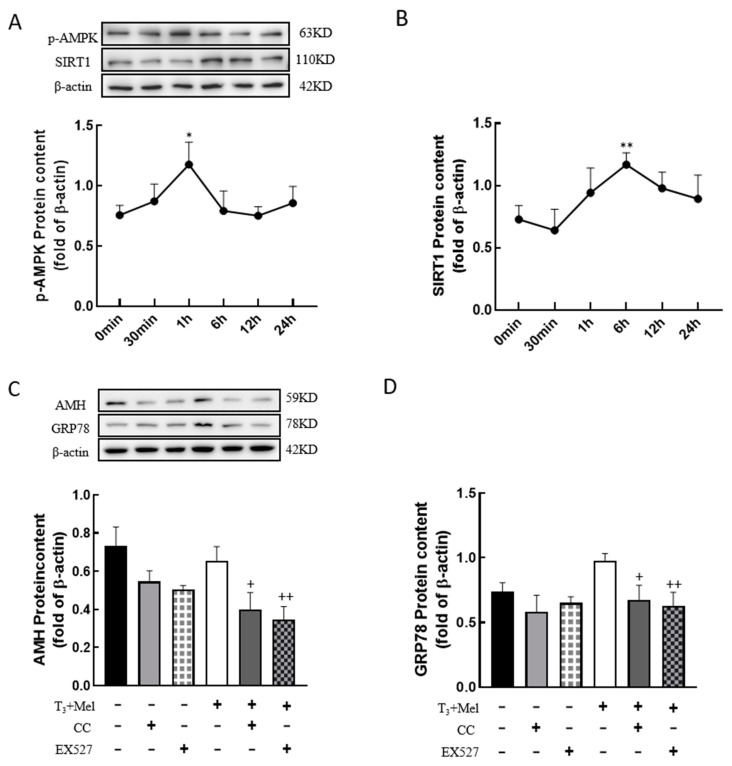
The AMPK/SIRT1 signaling pathway is involved in the regulation of granulosa cells via melatonin/T_3_. A/B protein was extracted at 0 min, 30 min, 1 h, 6 h, 12 h and 24 h after melatonin/T_3_ co-treatment. The protein expression of AMPK (**A**), SIRT1 (**B**), AMH (**C**) and GRP78 (**D**) was determined by Western blotting. * *p* < 0.05, ** *p* < 0.01, compared with 0 min; ^+^ *p* < 0.05, ^++^ *p* < 0.01, compared with melatonin/T_3_.

## Data Availability

The data and materials that support the findings of this study are available in the methods of this article.

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
