# Peer review of "Effects of Melatonin and 3,5,3′-Triiodothyronine on the Development of Rat Granulosa Cells"

_nutrients, 2024, doi:10.3390/nu16183085_

Round 1

Reviewer 1 Report

Comments and Suggestions for Authors

The problem presented in the manuscript is interesting but not novel.

1.The aim of the study -needs to be more specific because melatonin and T3 are not a part of diet or supplements

2.M&M parts must be written with detail: a/how the animals were killed;b/how many animals were used; 3.how many granulosa cells were used;d/how the culture was prepare;e/how many follicles were used?

3.Description of the results should be more clear

4. Discussion is chaotic

5.Conclusions: need to be written closely related to the results.

Reviewer 2 Report

Comments and Suggestions for Authors

Review on Nutrients manuscript by Wu et al., ... and Cheng Zhang

The manuscript by Wu et al. investigates a timely and important topic, the influence and protection from oxidative stress of rat granulosa cells. The methods are sound, the raw data were provided beforehand and are of good quality.

However, there are some issues depicted below that should be dealt with. It may also be helpful if a graphic depicting the supposed signal transduction pathways would be added.

General points

The authors may explain why oxidative stress may be relevant for rat granulosa cells and why they use H2O2 an not other oxidative stressors like NO. 

Also, what is the rationale behind using 100µM which inhibits cell viability just mildly (15-20%)? 

Is a direct, chemical interaction between H2O2 and melatonin/T3 possible and part of the resulting changes shown?

Moreover, the role of melatonin as a physiologically relevant antioxidant is under debate (Boutin and Jockers 2020).

The broadly accepted action of melatonin via its G-protein coupled receptors, MT1 and MT2, the authors may comment of this as an alternative way (other than antioxidative) to explain their effects (Shao et al.2024). The Melatonin receptor antagonist luzindole or other selective melatonin receptor antagonists may be helpful to distinguish between receptor-mediated and none-receptor mediated melatonin effects.

Melatonin and T3 are both hormones known to be involved in seasonal reproduction, among others acting on the granulosa cells of the ovary. If laboratory rats are really seasonal is debatable, but at least this should be discussed (Boutin and Jockers 2020; Hazlerigg et al. 2024)

Major points

Melatonin and T3 as hormones should show dose-dependent effects on their own. It would be nice if dose curves would be provided for both melatonin and T3 alone on ROS production to better understand if this is a physiological or pharmacological effect (Boutin and Jockers 2020). The combination of cause should be shown at the point where both hormones show their maximal effects (which may well be 100nM for Melatonin and 1nM for T3, but who knows?).

Minor points

Page 1 line 21-25 please clarify what you want to state, to be the sentence is hard to understand:

The results showed that melatonin alone or combined with T3 treated H2O2-induced granulosa cells, ROS decreased, cell proliferation and viability increased, cell apoptosis decreased, the expression of GRP78 was promoted, the expression of CHOP and Caspase-3 was inhibited, the expression of AMH was also promoted, and the expression of P16 was inhibited.“ 

You may try to separate into different sentences what melatonin and T3 (+Mel/T3 combined) do to granulosa cells with and without H2O2.

Page 2, Materials and Methods:

Source, purity (contaminants) and solute for melatonin may play an important role. The sources of melatonin and T3 are mentioned (Sigma-Aldrich, MO, USA) but not which preparations were used (batches, purity, etc.). Prominent contaminants in melatonin preparation are, among others, N-acetyl-serotonin, serotonin and variants thereof. Some of the latter substances may have effects of their own, especially at concentrations of the melatonin preparation at 100nM (0.1µM) and higher. The authors state that they solve melatonin in DMSO instead of the more commonly used solute ethanol. Any reason for that?

Page 3, line 117: AMPK and/or SIRT1 inhibitors were applied at 10 µM (which is quite common, but also rather high). Were the controls adjusted to the solvent content of these preparations?

Page 9 of 15, line 252, should read blotting

References

Boutin J. A., Jockers R. (2020) Melatonin controversies, an update. J. Pineal Res., e12702.

Hazlerigg D. G., Simonneaux V., Dardente H. (2024) Melatonin and Seasonal Synchrony in Mammals. J. Pineal Res. 76, e12996.

Shao R., Wang Y., He C., Chen L. (2024) Melatonin and its Emerging Physiological Role in Reproduction: A Review and Update. Curr. Mol. Med. 24, 449–456.

Round 2

Reviewer 1 Report

Comments and Suggestions for Authors

No further comments

Reviewer 2 Report

Comments and Suggestions for Authors

The authors have adequately responded to all criticism and suggestions.